# Amphibian-Friendly Water Drainages for Agricultural Landscapes, Based on Multiple Species Surveys and Behavioural Trials for *Pelophylax nigromaculatus*

Sanghong Yu [1,†], Yoonhyuk Bae [2,3,†], Yoonjung Choi [3], Daeun Yu [4], Yikweon Jang [2] and Amaël Borzée [3,*]

[1]   North Jeolla Province Institute of Science Education, Iksan 54549, Korea; partiii2002@naver.com
[2]   Division of EcoScience and Department of Life Science, Ewha Womans University, Seoul 03760, Korea; gyyh0303@gmail.com (Y.B.); jangy@ewha.ac.kr (Y.J.)
[3]   Laboratory of Animal Behaviour and Conservation, College of Biology and the Environment, Nanjing Forestry University, Nanjing 210037, China; partiii@naver.com
[4]   Department of Biotechnology, Pai Chai University, Daejeon 35345, Korea; partiii0724@naver.com
*   Correspondence: amaelborzee@gmail.com
†   These authors contributed equally to this work.

**Abstract:** Amphibians are the most threatened vertebrate group on earth, and one of the reasons for their decline is habitat loss. While some populations have persisted in agricultural wetlands such as rice paddies, the current anthropisation of landscapes is dealing a new blow to the survival of these species. In rice paddies, the new threats are especially visible through the increased channelization of water bodies with increasingly efficient drainage ditches, which become deadly traps. We first conducted surveys over three years to determine the use of ditches by frog species for natural versus concrete ditches, and thus relate to habitat adequacy as well as the probability of becoming trapped in concrete ditches. We then set up four types of experimental arena for escape trials. Experiments were replicated for the Black-spotted pond frog (*Pelophylax nigromaculatus*), as a proxy for other species abundant in rice paddies in the Republic of Korea. We determined that a slope of at least 70 degrees, with engraved patterns, was the only set-up from which frogs managed to escape. We recommend the implementation of this type of device in areas where a high concentration of animals is trapped, before phasing out the ancient design and relying on amphibian-friendly drainage ditches as they also support higher amphibian biodiversity.

**Keywords:** escape device; agricultural drainage; frogs; conservation; anuran

## 1. Introduction

Biodiversity is rapidly declining globally because of human activities [1] and especially because of habitat modification [2]. One such modification is that of agricultural landscapes, becoming increasingly covered in impermeable surfaces [3]. In addition, increased water drainage for agricultural purposes through the multiplication of drainage ditches is associated with amphibian decline [4]. Agricultural wetlands often originate from heavily modified natural wetland, and thus in habitats with high amphibian biodiversity [5]. The continuous transformation of wetlands since human development has negatively impacted amphibian communities, and especially during the agricultural revolution when drainage systems were multiplied. Currently, agricultural wetlands such as rice paddies provide substitute habitats to some amphibian species [6–9], and they have become of critical importance in some areas [10,11]. While imperfect, the value of agricultural wetlands for amphibian conservation is not negligible, and the decline in the area dedicated to rice agriculture in countries that were traditionally providing most of the rice is an additional stress to amphibian populations.

Despite their agricultural origins, drainage ditches in their natural form have numerous ecological functions [12]. For instance, they recycle nutrients and provide nutrients to land, while they also moderate the effects of herbicides [12]. Furthermore, natural drainage ditches host biodiversity, and especially amphibian species [7,13–15]. However, when they are not made of natural materials, ditches have a negative effect on biodiversity and populations [3], and especially when they are made out of smooth concrete and given a rectangular structure [14]. As a result of the modernization of agricultural landscapes, biodiversity is dropping [16–18], and solutions to revert this trend are needed [19].

While a few species of amphibians can climb vertical non-porous drainage ditches [20], most are unable to, and some mitigation efforts in the form of gullies [21] and escape ramps can be seen in Korean agricultural landscapes (pers. obs.). However, most of these have been developed without consideration of amphibian behaviour and characteristics and they are therefore not widely used. Numerous amphibians breed in rice paddies in the Korean Peninsula [22], and most are under risk of drying or starving in drainage ditches if they fall into one. In addition, animal-friendly drainages are not only important from an ethical perspective in view of the threatened status of some of the species trapped [23,24], but they are also needed to develop conservation plans to curb the negative population dynamics. Here, following repeated observations of our focal species being unable to escape agricultural ditches, we conducted surveys to determine the use of concrete versus natural ditches as natural amphibian habitat—and the risk of becoming trapped. We then tested several types of drainage ditches to determine how amphibians can best escape. Based on the escape rate of *Pelophylax nigromaculatus* in four different ditches structures, we define adequate angles and engraved patterns for easy escape by anurans.

## 2. Materials and Methods

### 2.1. Study Species and Research Area

*Pelophylax nigromaculatus* ranges from the Russian Far East to central China and Japan, through the Korean Peninsula [25]. The species is present in various types of habitat and they start breeding in early April in rice paddies, ditches and ponds [22].

### 2.2. Monitoring

Following repeated observations of anurans unable to escape from concrete ditches while relying on agricultural ditches as breeding habitat, we conducted surveys to determine which anuran species were most likely to become trapped in agricultural concrete ditches. We recorded the absence and presence of *P. nigromaculatus*, *P. chosenicus*, *Lithobates catesbeianus*, *Dryophytes japonicus* and *Rana* sp. in agricultural ditches between May and June, from 2014 to 2016. We conducted surveys between 3 p.m. and 4 p.m. at five locations once every second week in North Jeolla province in the Republic of Korea (35.9379° N 126.9929° E; 35.9242° N 126.9825° E; 35.9779° N 126.9749° E; 35.9850° N 126.9651° E; 36.0019° N 126.9617° E). At each location, we surveyed 50 m of both a modern concrete ditch and a natural ditch. We also measured temperature (°C) and humidity (%) at each survey point (HT-350 thermo-hygrometer; Iondo; Seoul, Republic of Korea). Frog sampling and housing.

As our surveys highlighted the presence of numerous *P. nigromaculatus* unable to escape ditches, we conducted manipulative experiments to test for the species ability to escape from different type of ditches. We used this species as a proxy for anurans in general. Twelve *P. nigromaculatus* were collected with a hand net from ditches in Wanju-gun, North Jeolla province (35.920569° 127.150166°) on 2 June 2018. Upon capture, we measured their weight down to 0.1 g (micro-electronic scale, HS-210/210, Hansung; Hwaseong, Republic of Korea) and front legs, rear legs and snout-vent length (SVL) down to 0.1 cm (Vernier calliper; NA505–150S; 0–150 mm, Bluebird; Seoul, Republic of Korea).

Each individual was placed by in separate plastic tank (23 × 15 × 16.5 cm) and fed two crickets (*Gryllus bimaculatus*; between 0.5 cm and 0.8 cm in length) every day during the duration of the experiment. Trials were conducted over the course of a single week and all individuals were release at the point of capture once tested.

*2.3. Experimental Models*

To test *P. nigromaculatus* ability to escape, we prepared four types of ditch replicates, with different angles and engraved patterns (Figure 1). For all types of experimental ditches, one of the sides was set vertically (90° angle with bottom) and without any engraved pattern, while the opposite side varied according to the type of replicate. When present, the engraved patterns were 1.5 cm-deep horizontal grooves engraved every 1 cm. The four treatments were: (1) 90° angle and no engraved patterns, (2) 90° angle with engraved patterns, (3) 70° angle towards the external side and no engraved patterns and (4) 70° angle towards the external side and engraved patterns. The four types of ditches were used as experimental arenas, measuring—height: 90 cm, width when 90° angle: 40 cm, width at top when 70°angle: 69 cm, ditch length: 180 cm, thickness: 10 cm (Figure 1). Both lateral sides were blocked with opaque acrylic panels. All ditches were made with using Styrofoam (compacted, Saehan Styrofoam, Yongin, Korea) supports covered with 1 cm of cement (mixing ratio: 5 to 7 L of water with soil and cement; 40 kg, Hanil, Jeonju, Korea) to adequately mimic ditches. We used HJ-Iso pink 300 (Brother Bond, Paju, Korea) and HJ-Iso pink 1800 (Brother Bond, Paju, Korea) to glue Styrofoam and cement.

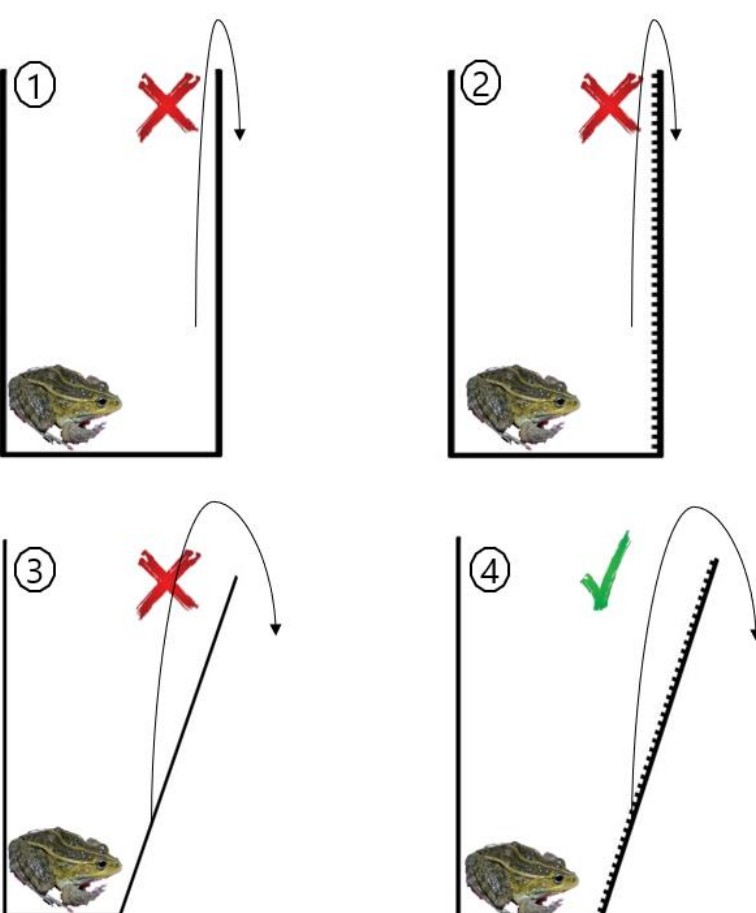

**Figure 1.** Experimental design for the escape trials. We used for ditch types: (**1**) 90° angle and no engraved patterns, (**2**) 90° angle with engraved patterns, (**3**) 70° angle towards the external side and no engraved patterns and (**4**) 70° angle towards the external side and engraved patterns. We used *Pelophylax nigromaculatus* for the trials, as a representative species for anurans in general.

### 2.4. Manipulative Experiments

For each escape trial, we placed a single adult *P. nigromaculatus* individual at the centre of the selected experimental ditch and observed escape outcomes within five min for each replicate (with escape or failure binary encoded). All of the 12 adult frogs were used for each experiment, with each individual tested once for each ditch type, in a random order. All experiments were filmed (TG 870, Olympus; Seoul, Republic of Korea) and escape outcome extracted from the videos not to disturb the frog during the experiment.

### 2.5. Statistical Analyses

Field monitoring. Prior to the data analysis, we tested for correlation between variables, and excluded the variable date from further analyses as it was significantly correlated (Pearson correlation test; $n = 60$) with weather ($r = 0.71$, $p < 0.001$), precipitation ($r = 0.64$, $p < 0.001$) and temperature ($r = 0.31$, $p = 0.018$). Weather was also removed from subsequent analyses because of the significant correlation with precipitation ($r = 0.99$, $p < 0.001$) and humidity ($r = 0.43$, $p = 0.001$). Finally, temperature was removed because of the correlation with precipitation ($r = 0.33$, $p < 0.011$) and humidity ($r = 0.44$, $p < 0.001$).

To test for the presence or absence of the five amphibian species in concrete vs. natural ditches (binary encoded), we used a binary logistic regression with ditch type as dependent variable, and time precipitation, humidity, *P. nigromaculatus*, *P. chosenicus*, *D. japonicus*, *Rana sp.* and *L. catesbeianus* as response variables in the model. These species were selected based on the results of the surveys. A backward conditional model was selected to assess the impact of each variable on the model. We visually tested for the absence of outliers by analysing box-plots, we determined the normal distribution of the data with the Kolmogorov–Smirnov test for normality with the Lilliefors significance correction ($0.15 \leq D(60) \leq 0.38$, $p < 0.002$), and determined the homogeneity of variance with Levene's test for homogeneity of variances. The variables were significantly homogeneous ($3.11 < F(1,58) < 13.02$, $p < 0.005$) at the exception of humidity: $F(1,58) = 0.14$, $p = 0.708$. We also detected an outlier for precipitation but decided it would not affect the model.

Manipulative experiments. Among the covariates collected, only frog ID and hind-leg length were not significantly correlated (Pearson correlation; $n = 48$, $r = 0.28$, $p = 0.550$), and consequently the variables front leg length, SVL and weight were removed from all subsequent analyses. We used a binomial logistic regression to discriminate the impact of the four types of ditches on the frog ability to escape, with escape as dependent variable, and ditch type, frog ID and hind-leg length as response variables. However, because of the complete separation in the data (i.e., all the frogs from one ditch type managed to escape but no frog escaped for any of the other ditch types), the log-likelihood values could not be calculated. All statistical analyses were conducted in SPSS v 21.0 (SPSS, Inc., Chicago, IL, USA).

## 3. Results

### 3.1. Monitoring

We highlighted the presence of *Pelophylax chosenicus*, *P. nigromaculatus*, *Dryophytes japonicus*, *Rana* sp. and *Lithobates catesbeianus* during our surveys A larger number of individuals was found in natural ditches compared to concrete ditches for all species with the exception of *Rana* sp., which was not found in natural ditches (Table 1; Figure 2). The results of the binary logistic regression supported these results for all species, with the exception of *Lithobates catesbeianus* (Table 2). The analysis resulted in three steps, all significant under Omnibus tests of model coefficients (Step 1: $\chi^2 = 32.01$, df = 8, $p < 0.001$; Step 2: $\chi^2 = 31.59$, df = 7, $p < 0.001$; Step 3: $\chi^2 = 30.20$, df = 6, $p < 0.001$), and explaining 55.1, 54.6 and 52.7 % (Nagelkerke pseudo-R2) of the variance, respectively. The presence of *Rana* sp. and *D. japonicus* were significantly different between ditch types for all three steps, while it was only marginally significant for *P. nigromaculatus* in step one, before reaching significance in step 2 and 3 (Table 2).

**Table 1.** Descriptive statistics for species occurrence during field surveys. The surveys were conducted in North Jeolla Province, Republic of Korea, between May and June from 2014 to 2016. The data are split between concrete drainage ditches and natural ditches in the vicinity of rice paddies.

| Species | Concrete Ditch | | Natural Ditch | |
|---|---|---|---|---|
| | **Mean** | **SD** | **Mean** | **SD** |
| *Pelophylax chosenicus* | 6.50 | 4.12 | 7.16 | 2.36 |
| *Pelophylax nigromaculatus* | 1.31 | 0.79 | 2.18 | 1.48 |
| *Dryophytes japonicus* | 6.63 | 3.74 | 8.64 | 2.17 |
| *Rana* sp. | 0.13 | 0.50 | 0.00 | 0.00 |
| *Lithobates catesbeianus* | 1.06 | 0.44 | 1.23 | 0.89 |

**Table 2.** Results of the Binary Logistic Regression used to test for the presence or absence of the five amphibian species in concrete vs. natural ditches. The five species were *Pelophylax nigromaculatus*, *P. chosenicus*, *Dryophytes japonicus*, *Rana* sp. and *Lithobates catesbeianus* and the surveys were conducted in central Korea in spring 2017.

| | | **B** | **S.E.** | **Wald** | **df** | *p*-**Value** |
|---|---|---|---|---|---|---|
| | Precipitation (mm) | −1.39 | 0.83 | 2.83 | 1 | 0.092 |
| | Humidity | −0.05 | 0.03 | 2.55 | 1 | 0.111 |
| | *P. chosenicus* | −0.30 | 0.18 | 2.77 | 1 | 0.053 |
| Step 1 | *P. nigromaculatus* | 0.48 | 0.38 | 1.56 | 1 | 0.173 |
| | *D. japonicus* | 0.43 | 0.19 | 5.08 | 1 | **0.010** |
| | *Rana* sp. | −11.94 | 20,096.49 | 0.00 | 1 | **0.019** |
| | *L. catesbeianus* | 0.32 | 0.50 | 0.41 | 1 | 0.519 |
| | Time | 0.00 | 0.00 | 0.82 | 1 | 0.365 |
| | Precipitation (mm) | −1.50 | 0.80 | 3.46 | 1 | 0.063 |
| | Humidity | −0.05 | 0.03 | 2.98 | 1 | 0.084 |
| | *P. chosenicus* | −0.31 | 0.18 | 3.05 | 1 | **0.048** |
| Step 2 | *P. nigromaculatus* | 0.37 | 0.33 | 1.29 | 1 | 0.233 |
| | *D. japonicus* | 0.44 | 0.19 | 5.63 | 1 | **0.006** |
| | *Rana* sp. | −11.99 | 20,096.49 | 0.00 | 1 | **0.017** |
| | Time | 0.00 | 0.00 | 1.77 | 1 | 0.183 |
| | Precipitation (mm) | −1.62 | 0.77 | 4.39 | 1 | 0.036 |
| | Humidity | −0.05 | 0.03 | 3.57 | 1 | 0.059 |
| Step 3 | *P. chosenicus* | −0.31 | 0.17 | 3.11 | 1 | **0.045** |
| | *D. japonicus* | 0.48 | 0.19 | 6.83 | 1 | **0.002** |
| | *Rana* sp. | −12.04 | 20,096.49 | 0.00 | 1 | **0.016** |
| | Time | 0.00 | 0.00 | 2.74 | 1 | 0.098 |

*3.2. Manipulative Experiments*

The only experimental setting from which the frogs managed to escape was the one with the 70-degree inclination and engraved patterns. All frogs from this experiment managed to escape within the time of the experiment, but no frog managed to escape from any other ditch type (Figure 1). The binomial logistic regression used to discriminate the impact of the four types of ditches on the frog ability to escape was an adequate fit for the model ($\chi^2 = 53.98$, df = 5, $p < 0.001$) and explained 100 % of the variance (Nagelkerke pseudo-R2). The log-likelihood could not be calculated for the covariates (frog ID SD = 1740.52; hind-leg length SD = 1316.97), but there were no significant differences for the covariates between treatments ($p > 0.999$). The escape capabilities were however significantly different between ditch types (log-likelihood = 54.08, df = 4, $p < 0.001$).

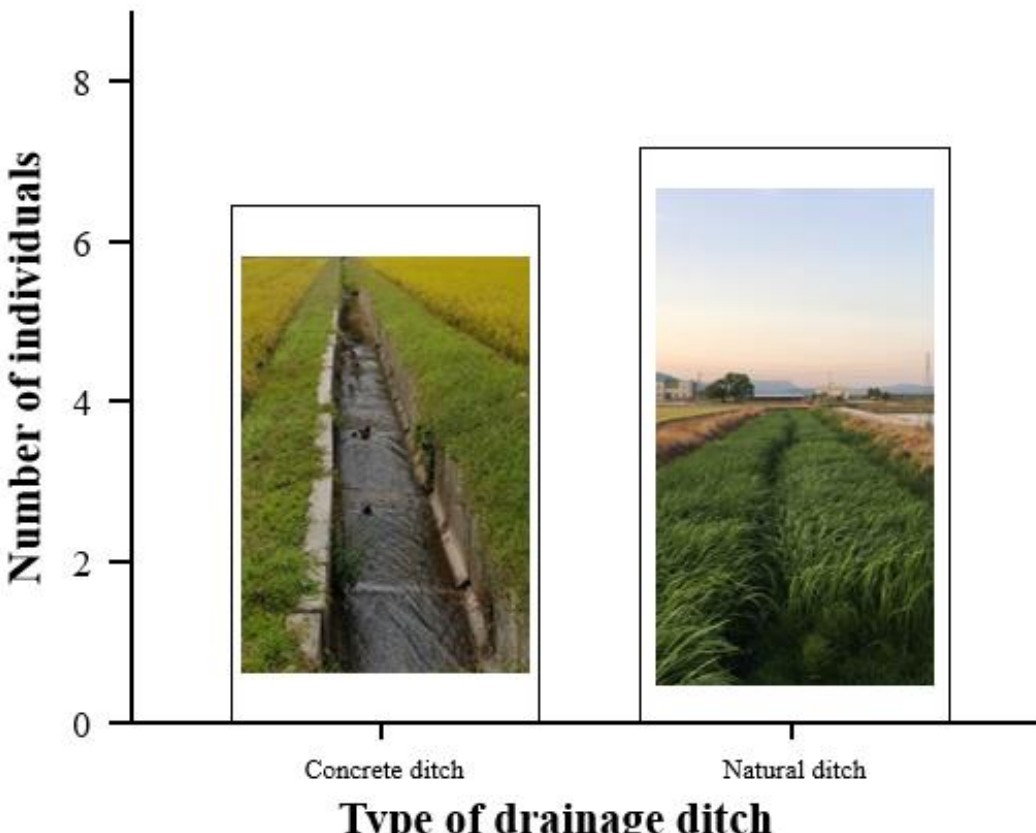

**Figure 2.** Mix-species anuran assemblage encountered by ditch type. We recorded the absence and presence of *Pelophylax nigromaculatus*, *P. chosenicus*, *Lithobates catesbeianus*, *Dryophytes japonicus* and *Rana sp.* in agricultural ditches in North Jeolla province, Republic of Korea, between May and June, from 2014 to 2016.

## 4. Discussion

Our results first showed that most amphibian species breeding in rice paddies are using the ditch habitat in early spring. Generally, all species became trapped in concrete ditches set up for irrigation, and potentially for long periods as *Rana* sp. are not generally present in rice paddies after breeding in early winter. Our results are in line with the literature available, where many amphibians are becoming trapped in ditches with 90° angles between the bottom and vertical edges [26]. Our second set of results showed that only one of the experimental ditch types tested was efficient to let amphibians escape.

While some species such as treefrogs have suction cups that generally allow them to escape from vertical structures [27], the latest type of concrete ditch used in the Republic of Korea is too smooth for treefrogs to escape (pers. obs. for *Dryophytes japonicus* and *D. suweonensis*). Other species such as *Pelophylax chosenicus* and *P. nigromaculatus* cannot escape from ditches with a 90° structure, nor from those at 70° in the absence of engraved patterns. We recommend this type of drainage ditch over one with a lower angle due to space restriction in agricultural wetlands, and the maintenance of water flows for irrigation purposes.

Our results also show the presence of a larger number of *P. chosenicus* individuals in natural ditches compared to concrete ditches, likely due to microhabitat segregation between *Pelophylax* species when in syntopy [28,29]. The recent decline in the species has been linked to modern agriculture [30], and modernization of the landscape, such as concrete ditches, certainly has a role to play in this decline. Specifically, *P. chosenicus* is more generally found in vegetated habitat, which now translates into agricultural wetlands [31,32] as most of the natural wetlands have now been converted [33]. Natural ditches are suitable as an alternative natural habitat for *P. chosenicus* and habitat restoration will improve the

ecological function of manmade wetlands, despite their limitation [4]. In addition, it is also interesting to see that dispersion of *Lithobates catesbeianus* across agricultural wetlands in Korea [34] is a pattern followed by numerous invasive amphibian species [35,36], and the development of agricultural ditches is likely to have helped in the dispersion of the species.

## 5. Conclusions

The presence of several amphibian species in concrete ditches highlights that water drainage systems have the potential to become deadly traps for all anuran species of the area. We recommend the implementation of the type of ditch demonstrated here, with our focal species, but also in areas where the threatened *P. chosenicus* is present as the use of this type of ditch will enable reaching much needed conservation targets. An even better option would be to switch back to natural drainage ditches as they have the potential to become important conservation tools when used properly [12,37].

**Author Contributions:** Conceptualization, S.Y., Y.C. and D.Y.; methodology, S.Y., Y.J. and A.B.; formal analysis, Y.B. and A.B.; data curation, S.Y. and Y.B.; writing—original draft preparation, S.Y. and Y.B.; writing—review and editing, Y.J. and A.B. All authors have read and agreed to the published version of the manuscript.

**Funding:** This research was funded by the Foreign Youth Talent Program from the Ministry of Science and Technology of the People's Republic of China, grant number QN2021014013L to AB.

**Institutional Review Board Statement:** The study was conducted in accordance with the Ethics Committee of Nanjing Forestry University (202-20-14).

**Data Availability Statement:** Not applicable.

**Acknowledgments:** We are thankful to the program Love the Earth Explorers from Donga Science for their support during the project.

**Conflicts of Interest:** The authors declare no conflict of interest.

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
