# Peer review of "Amphibian-Friendly Water Drainages for Agricultural Landscapes, Based on Multiple Species Surveys and Behavioural Trials for Pelophylax nigromaculatus"

_diversity, doi:10.3390/d14050414_

Round 1

Reviewer 1 Report

Dear Authors,

Please find my suggestions below.

Line 14 more like vertebrates or vertebrate group.

Line 140 scientific names are not italics here, correct it.

Line 155 change to variables.

Line 174-175 this sentence more belongs to the discussion. Consider to delete or remove it.

Line 181 Figure 2. It looks nice that you provided a picture for the concrete ditch but please insert another one for the natural too.

This research is very interesting and timely as these ditches would serve as rather good habitats worldwide but instead they are frequently paved and thus turning into traps. As you have the knowledge now how to perform such analyses please consider to repeat it with other species and for much longer time periods such as one day or days.

Author Response

Reviewer 1:

è Thank you for the time spend reviewing our manuscript. We have answered to all comments in detail below.

Please find my suggestions below.

Line 14 more like vertebrates or vertebrate group.

è Corrected as suggested: “Amphibians are the most threatened vertebrate group on earth, and one of the reasons for their decline is habitat loss”.

Line 140 scientific names are not italics here, correct it.

è Corrected as suggested

Line 155 change to variables.

è Corrected as suggested

Line 174-175 this sentence more belongs to the discussion. Consider to delete or remove it.

è We have deleted this sentence, and moved it to the conclusion

Line 181 Figure 2. It looks nice that you provided a picture for the concrete ditch but please insert another one for the natural too.

è We have modified the figure as suggested, please see the newly submitted manuscript.

This research is very interesting and timely as these ditches would serve as rather good habitats worldwide but instead they are frequently paved and thus turning into traps. As you have the knowledge now how to perform such analyses please consider to repeat it with other species and for much longer time periods such as one day or days.

è Thank you for the positive feedback, and will strive to expand as recommended in the future.

Reviewer 2 Report

  1. The authors presented the results of their experiments. These experiments were done with one species of amphibian. It seems to me that because of this it is necessary to change the title of the manuscript.
  2. This manuscript is a brief report. Therefore, in the introduction it is worth specifying more specific publications on the decline of amphibian biodiversity. For example, such publications (https://dx.doi.org/10.24189/ncr.2021.033 ; https://dx.doi.org/10.24189/ncr.2019.036 ), and others.
  3. Line 215-217. Why does P. chosenicus prefer natural ditches and is found in them in greater numbers?
  4. Line 227-229. Your recommendations should be based on the experiments conducted. You have conducted experiments only with P. nigromaculatus, but not with P. chosenicus. Or justify your recommendations.
  5. What assumptions can the authors have about the possible use of drainage ditches with inclined sides for other amphibians?
  6. There is no conclusion. It is necessary to write a conclusion.

Author Response

Reviewer 2:

è Thank you for the time spend reviewing our manuscript. We have answered to all comments in detail below.

The authors presented the results of their experiments. These experiments were done with one species of amphibian. It seems to me that because of this it is necessary to change the title of the manuscript.

è We agree with the reviewer, our experiments were conducted for a single species of amphibian. We are not certain about the correction suggested here. Out of deduction (as it is amphibian from the experiments and agricultural from the surveys) we suspect the comment is about water drainage. This is a neutral word that we think fit the context, but we will follow other recommendations if requested.

This manuscript is a brief report. Therefore, in the introduction it is worth specifying more specific publications on the decline of amphibian biodiversity. For example, such publications (https://dx.doi.org/10.24189/ncr.2021.033 ; https://dx.doi.org/10.24189/ncr.2019.036 ), and others.

è We agree with the reviewer that specific examples are important, however the first publication recommended is about the high survival rate of a newt species, and the second one is about threat assessment. Maybe the links pasted were inadvertently inverted?

Line 215-217. Why does P. chosenicus prefer natural ditches and is found in them in greater numbers?

è The two species have segregated ecological requirements, and these may be further reinforced in in syntopy. We have added this explanation such as: “Our results also show the presence of a larger number of P. chosenicus individuals in natural ditches compared to concrete ditches, likely due to microhabitat segregation be-tween Pelophylax species when in syntopy [28,29]”.

Line 227-229. Your recommendations should be based on the experiments conducted. You have conducted experiments only with P. nigromaculatus, but not with P. chosenicus. Or justify your recommendations.

è For ethical reasons, and because P. chosenicus is threatened, it is not possible to test P. chosenicus, but the physiology of the two species is similar enough to expect commonalities. We have added a sentence to tone down the recommendations (and moved it to the conclusion): “The presence of several amphibian species in concrete ditches highlights that water drainage systems have the potential to become deadly traps for all anuran species of the area. We recommend the implementation of the type of ditch demonstrated here, with our focal species, but also in areas where the threatened P. chosenicus is present as the use of this type of ditch will enable reaching much needed conservation targets”.

What assumptions can the authors have about the possible use of drainage ditches with inclined sides for other amphibians?

è We expect all species to be able to escape with our design, but as it is not demonstrated, and potentially misleading, we would prefer not to expand of unfounded expectations.

There is no conclusion. It is necessary to write a conclusion.

 è We have added a conclusion: “The presence of several amphibian species in concrete ditches highlights that water drainage systems have the potential to become deadly traps for all anuran species of the area. We recommend the implementation of the type of ditch demonstrated here, with our focal species, but also in areas where the threatened P. chosenicus is present as the use of this type of ditch will enable reaching much needed conservation targets. An even better option would be to switch back to natural drainage ditches as they have the potential to become important conservation tools when used properly [12,37]”.

Round 2

Reviewer 2 Report

Dear authors. Thank you for answering my questions. However, it seems to me that the name still needs to be changed. Your results and conclusions are based on experiments with one species of amphibian.

Author Response

Thank you for the additional feedback, we have updated the title such as: Amphibian-friendly water drainages for agricultural landscapes, based on multiple species surveys and behavioural trials for Pelophylax nigromaculatus